# Transverse Profiles of Belt Core Damage in the Analysis of the Correct Loading and Operation of Conveyors

Leszek Jurdziak [1,*], Ryszard Błażej [1], Agata Kirjanów-Błażej [2] and Aleksandra Rzeszowska [1]

1   Faculty of Geoengineering, Mining and Geology, Wroclaw University of Science and Technology, Na Grobli 15 St., 50-421 Wrocław, Poland; ryszard.blazej@pwr.edu.pl (R.B.); aleksandra.rzeszowska@pwr.edu.pl (A.R.)
2   Faculty of Information and Communication Technology, Wroclaw University of Science and Technology, Janiszewskiego 11/17 St., 50-372 Wrocław, Poland; agata.kirjanow-blazej@pwr.edu.pl
*   Correspondence: leszek.jurdziak@pwr.edu.pl

**Abstract:** This article presents an analysis of the transverse profile of belt damage in the context of the proper loading and operation of conveyors. The aim of this study was to identify and understand the characteristic features of damage and their placement that may occur in conveyor belts during operation and indicate abnormalities in any of the components of the conveyor system, especially during loading at chutes. A total of seven different conveyors were examined, and the obtained results allowed for a thorough comparison and investigation of the distribution of belt damage on their cross-sectional profiles. This article discusses factors that may contribute to the occurrence of unevenly distributed belt damage. The conclusions from the conducted research can be of significant importance for conveyor owners and operators as they enable an effective assessment of the conveyor belt's condition, the correctness of the conveyors, and the chutes' designs as well as the implementation of necessary design changes, correct actions, and repairs. Through proper belt monitoring and maintenance, the risk of failure can be minimized, extending the belts' lifespan and ensuring the efficiency of the transportation process. The article presents practical approaches to the analysis of the cross-sectional profile of damage, serving as a valuable source of information for individuals interested in optimizing the transportation process and maintaining efficient and safe conveyor operation.

**Keywords:** conveyor belt; damage distribution; NDT; diagnostic systems

## 1. Introduction

Belt conveyors play an incredibly important role in industry, serving as devices for efficient transportation of various materials with high capacity [1,2] over long distances [3]. They are utilized in numerous sectors such as mining, heavy industry, chemicals, ports, and more. Belt conveyors are capable of carrying large quantities of materials over substantial distances, enabling a continuous flow of bulk products. Properly designed belt conveyors can handle both light and heavy materials while maintaining high efficiency and low energy consumption, typically around 1 ton per kilometer [4–6]. They are engineered to minimize frictional resistance and energy consumption, thereby ensuring an energetically optimal continuous transport. By utilizing advanced drive technologies, high-strength belts, and sophisticated control systems, it is possible to transport large volumes of materials with minimal energy consumption, leading to reduced operational costs and improved overall production efficiency. Figure 1 illustrates a schematic diagram of a belt conveyor construction.

The use of continuous transportation, such as belt conveyors, also contributes to increased workplace safety by eliminating the need for manual handling of heavy loads, thereby reducing the risk of accidents and injuries to workers. Additionally, belt conveyors are often equipped with various monitoring systems to ensure proper operation and

provide warnings in case of detected hazards, further enhancing the safety of transport operations [7,8].

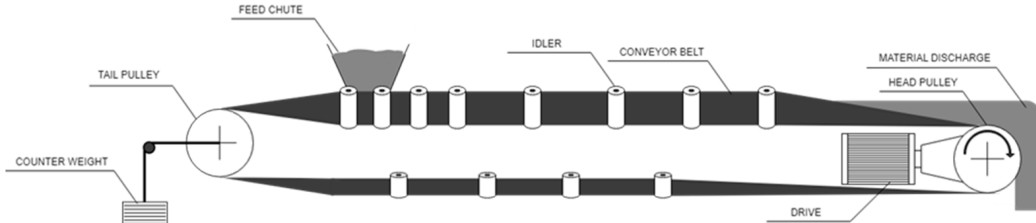

**Figure 1.** Belt conveyor construction diagram.

Belt conveyors are designed to meet specific and diverse industrial needs, taking into account the specific requirements of production, types of transported materials, and working conditions. This flexibility makes them a versatile solution that can be easily adapted to various industrial sectors. However, it is important to note that the precise design of the belt conveyor is crucial to ensure proper operation and minimize the wear of its components [9–11]. Even small errors in the design can lead to uneven wear of the belt, idlers, bearings, and other conveyor components. As a result, this can shorten the lifespan of the conveyor, increase the risk of failures, and require more frequent repairs or component replacements, thereby reducing their reliability [12].

To maintain the high efficiency and reliability of belt conveyors, measurement systems for monitoring and diagnosing their condition are extremely important [13–17]. With technological advancements, measurement systems enable increasingly precise monitoring and more comprehensive and accurate analysis of various conveyor operation parameters, allowing for the early detection of potential issues and the implementation of appropriate corrective actions without the need to halt the entire transport and production line. These measurement systems play a crucial role in ensuring optimal performance, minimizing downtime, and optimizing maintenance strategies, thereby enhancing the overall efficiency and reliability of belt conveyors.

In recent years, there has been increasing attention given to the development of diagnostic methods [18–21]. Automatic systems utilizing non-destructive testing methods without the need to stop the conveyors have gained popularity. Compared to traditional methods such as visual inspection, automatic non-destructive methods allow for faster and more precise assessment of the technical condition, leading to decisions regarding the need for repair or replacement of conveyor components well in advance, enabling preparation for these actions [22]. This shortens the time required to carry out the necessary maintenance or replacement tasks.

In the diagnostics of belt conveyors, various non-destructive methods are employed, enabling a detailed analysis of different conveyor components. These methods allow for the identification of failures in the belt core, such as corrosion, cracks, cuts, or missing links. The effectiveness and precision of the diagnostic methods are crucial, as they enable the early detection of faults and prevent more serious failures. By implementing effective diagnostic systems, prompt responses to changes in the technical condition of belt conveyors can be ensured, minimizing production downtimes, increasing the durability of conveyor components, and improving the efficiency of material transportation systems, as well as predicting remaining operational time and planning preventive maintenance stops.

At the Wrocław University of Science and Technology, a diagnostic system called DiagBelt has been developed for monitoring and diagnosing the technical condition of belt conveyor cores [19,20]. The system consists of a data analysis program and a measurement belt (provided by Beltscan System Pty, Southport, Queensland, Australia), which includes 112 coils placed at regular intervals of 25 mm. Additionally, there are two belts of permanent magnets located above and below the examined belt, spaced approximately 30 mm from its covers. Figure 2 shows a schematic cross-section of a conveyor belt with steel cords.

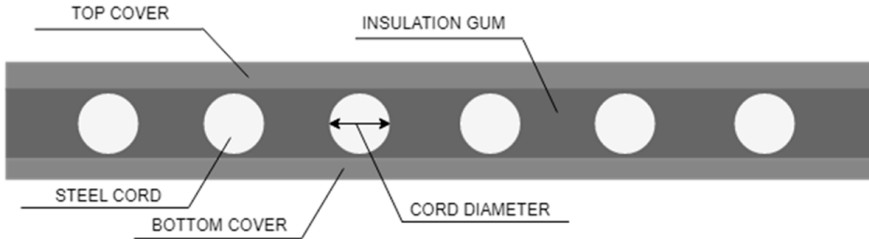

**Figure 2.** Schematic cross-section of a conveyor belt with steel cords.

Figure 3 presents the components of the DiagBelt+ diagnostic system, while Figure 4 shows the schematic of the device installation on the conveyor belt.

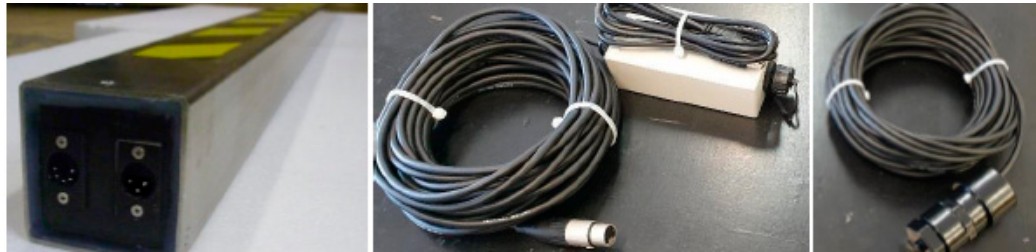

**Figure 3.** Components of the DiagBelt+ system (measuring head, data acquisition module with USB interface, tachometer).

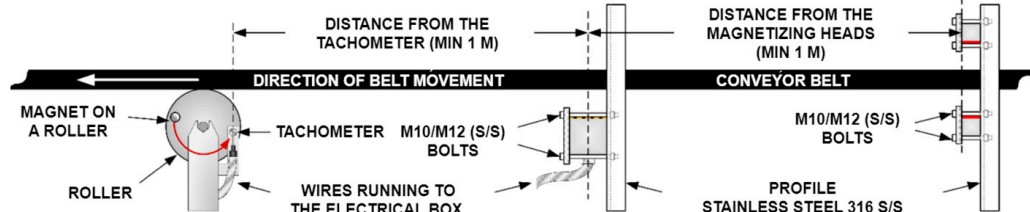

**Figure 4.** Schematic of the installation of the measuring head and magnetizing heads.

In Figure 5, the installation method of the DiagBelt system on a conveyor belt in the Belt Transport Laboratory of Wroclaw University of Science and Technology (LTT) is shown. The DiagBelt system can be mounted on both the upper and lower strands of the belt, providing flexibility and the ability to monitor the technical condition of the conveyor belt at various points along its path.

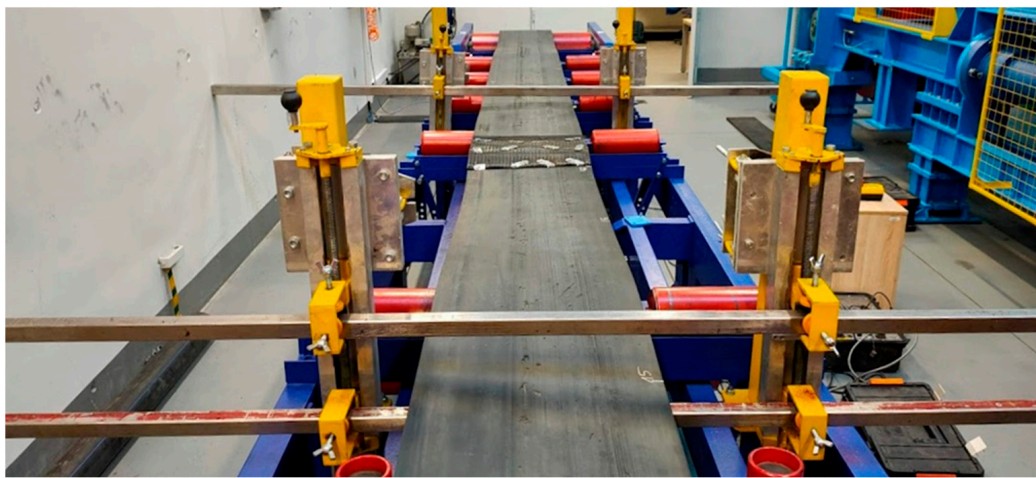

**Figure 5.** Installation of the DiagBelt system at the laboratory station in LTT.

The coils placed on the measurement head register changes in the magnetic field that occur at locations where discontinuities in the steel links of the belt are present (such as cuts,

missing links, or broken loops). This allows for the assessment of the technical condition of the belt core. The measurement signals from the coils are amplified and collected in a data recording module and then transmitted to a computer. The system allows for the adjustment of coil sensitivity and measurement frequency, enabling the customization of the measurements to individual requirements. The measurement results are saved in a CSV file, allowing for further analysis to assess the technical condition of the belt loop composed of multiple sections, as well as for planning the repair, maintenance, and replacement of worn-out sections and fragments.

Figure 6 presents an example of measurement data obtained from the DiagBelt system, visualized in the form of a two-dimensional image. This image was obtained by scanning a reference belt in the LTT laboratory, on which several types of failures were simulated. Such an image allows for a detailed analysis and identification of failures in the core of the conveyor belt.

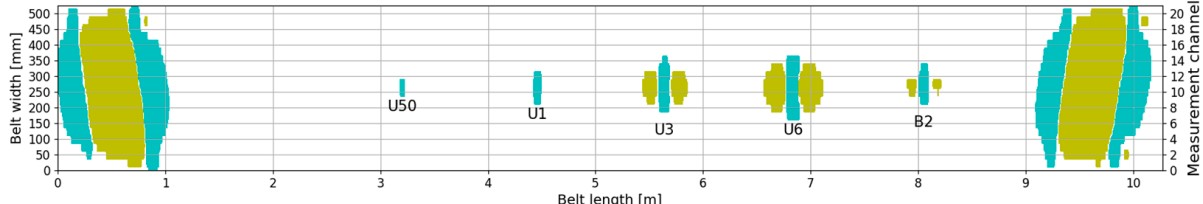

**Figure 6.** Example visualization of measurement data from the DiagBelt diagnostic system.

Two-dimensional image of simulated damage signal patterns (Figure 6) and their actual image on the test belt can be seen above (Figure 7).

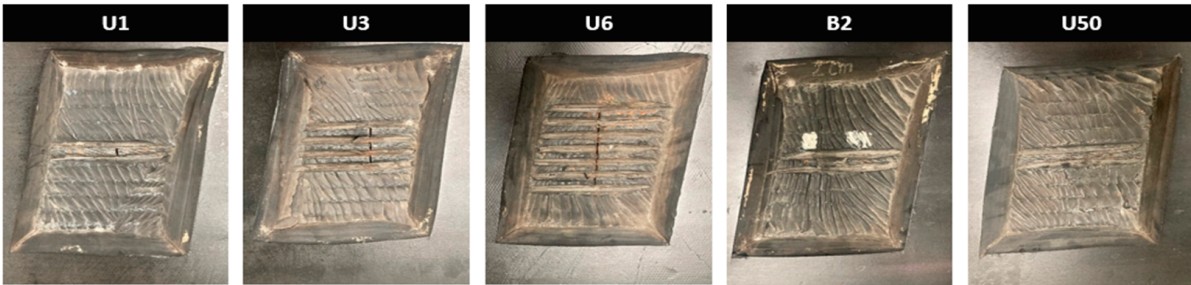

**Figure 7.** Actual appearance of simulated damage on the belt at LTT. U1—one cord cut; U3—three cords cut; U6—six cords cut; B2—2 cm missing cord; U50—50% reduction in cord cross-section.

The damage signal always consists of a core signal (blue color in the 2D visualization) and, depending on the size of the damage, it may also have one or two accompanying signals located before or after the core signal. For small failures (partial link damage, damage to one link), the damage signal consists only of a positive-polarity signal (core signal). In the case of larger failures (damage to three links), accompanying signals also appear around the core signal. However, it should be noted that the magnitude of the signal depends on the operating parameters of the conveyor belt and the measurement head—the belt speed, the distance between the head and the links on the belt, and the sensitivity threshold set for the coils in the recording head to detect magnetic signal changes [23].

Currently, in Polish mines, the use of core diagnostics utilizing magnetic and X-ray systems for assessing the condition of the core is very rare. Such devices have been implemented by a team from the Wroclaw University of Science and Technology in the Turów and Belchatów mines. Research has also been conducted by companies specializing in the magnetic testing of steel ropes in mining shafts and bridges [24]. These devices can identify failures, but they rely on one-dimensional signals from a narrow measurement head or several one-dimensional signals that represent aggregated damage data for a 40 cm wide belt. Therefore, it is difficult to observe the distribution of damages across the transverse section of the belt, as the signal is aggregated into a few points (e.g., five for a 2000 mm wide belt).

In the world, there are diagnostic systems available that provide a two-dimensional image of failures on the belt surface, even with high resolution. Examples include X-ray systems (Shanxi DMC Co., Ltd., Shanxi, China), CB Guard (VBG Conveyor Belt Gateway, Hamburg, Germany) or the magnetic system BeltGuard by BeltScan System Pty from Australia [25]. Other systems have lower resolution, resulting in the aggregation of failures into 10 cm wide or wider heads. They also do not offer the automated processing of data on identified failures. The application of X-ray imaging in diagnosing conveyor belts [25–27] can be deemed inefficient at times due to the laborious analysis of hundreds of resulting images to identify areas of damage. While this method accurately depicts the appearance of individual cables in the belt, interpreting the resulting images can be challenging.

The DiagBelt and DiagBelt+ systems developed at the Wrocław University of Science and Technology by the authors of the article utilize a measurement head from Beltscan with very high resolution (80 sensors per 2 m width, every 25 mm). The obtained scanning results, in the form of a digital map of magnetic signal variations, are further processed in specialized software, allowing for the determination of various statistics regarding the failures (e.g., failure measures). The system counts failures, classifies them according to size (dimensions of the rectangle spanned over the identified damage), and calculates the exact area of each damage (the identifying signal's irregular surface), records and sums elements describing the failures (location relative to the splices and belt edge, dimensions along the axis and transverse direction, sum of the rectangle's area, exact area) for the entire loop, sections between splices, and selected belt fragments. Additionally, the system determines the density of failures (number of failures per meter of belt) and the density of the failure area (failure area in $cm^2$ per meter of belt) and presents statistics along the belt axis (density histogram, area density) and transversely (distribution of failures (area) across the section). The presentation of the damage distribution across the section is a unique feature of the presented system since it is performed with a resolution of 25 mm per section and every 2.5 mm along the axis for each meter per second of belt speed. Therefore, the distribution represents the results of precise scanning (40 (which represents the number of measurement channels across 1 m belt width) multiplied by 400 (which represents the sampling frequency) and divided by v (which represents the belt velocity) belt width per square meter of belt). For instance, at a belt speed of 1 m/s, the measurement grid contains 16,000 data points per square meter of the belt. Similarly, at a belt speed of 6 m/s, the measurement grid contains 2666 data points per square meter of the belt.

The reliability and accuracy of the measurement allow for a precise determination of the belt wear intensity across the section (in terms of the number of failures per channel per month of usage for each channel), which can serve as a basis for determining the loading method of the material on the belt and its consequences.

## 2. Materials and Methods

The conducted study analyzed measurement data obtained from scanning conveyor belts using the DiagBelt system in mines in Poland. The performed scans allowed for the identification of damage locations on the belt surface. After removing all connections from the data, it was possible to analyze the frequency of damage on the belt (between the connections) on each of the measurement channels. This provided an image of the distribution of damage across the cross-section of the belt. The high density of coils (40 per meter of cross-section) allowed for visualizing the distribution of damage on 90 channels for a belt width of 2250 mm. Figure 8 presents an example image depicting the distribution of the number of failures across the cross-section of one of the studied belt loops.

The image presents a distribution of the number of failures composed of the main signal (blue bars) and accompanying signals (yellow bars). The numbers corresponding to the blue bars illustrate the count of failures, while the yellow bars represent two or one accompanying signals carrying information about the type of damage (e.g., line breakage, cross-sectional loss, etc.). It can be observed that the distribution is trimodal, with the central area of the belt exhibiting a noticeably higher number of failures compared to the

other two local maxima around channels 26–30 and 70. There are also a few failures at the edges. The uneven distribution of failures is attributed to the design of the conveyor's loading and support system, as well as the intensity of material impact (e.g., overburden) at different locations across the belt's width. The failures at both edges likely resulted from edge wear and corrosion. The areas of the belt practically free from failures (near the upper edges of the lateral three-roll idler set in the shape of a basin) indicate that hardly any particles fall there, and their oblique arrangement likely increases the impact energy of particles that could damage the core. The uneven distribution of failures accelerates belt wear, as the most damaged (and likely most worn) sections will determine the need for replacement.

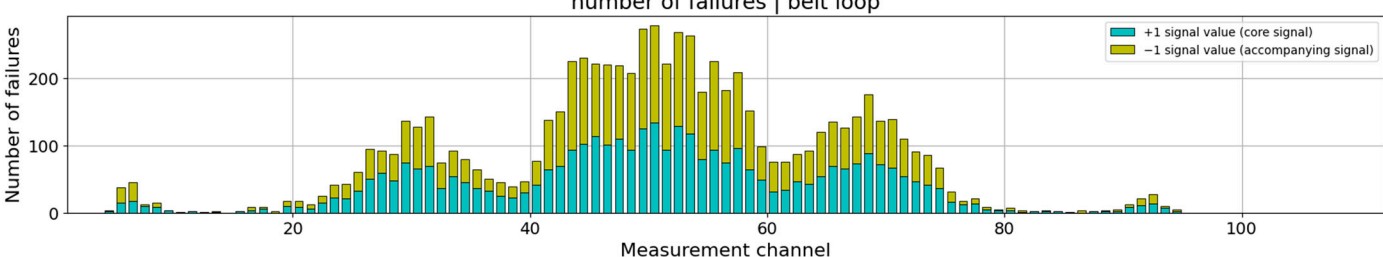

**Figure 8.** Example histogram of damage distribution across the width of the conveyor belt (Conveyor A).

In addition to studying the distribution of failures across the cross-section, it is also valuable to analyze other measures of failures, such as investigating the extent of failures in the longitudinal and transverse directions. Extent refers to an aggregated measure in the form of the sum of sizes of all failures in the longitudinal and transverse directions (Equations (1) and (2)).

$$E_X = \sum_{i=1}^{N} l_i, \tag{1}$$

$$E_Y = \sum_{i=1}^{N} w_i, \tag{2}$$

where:

- $E_X$—longitudinal extent;
- $E_Y$—transverse extent;
- $N$—number of failures in the section;
- $l_i$—length of the *i*-th damage in millimeters;
- $w_i$—width of the *i*-th damage in millimeters.

To account for the varying length of conveyor belts, the extent of failures can be averaged per meter of the belt, resulting in the density of longitudinal/transverse extent. This approach considers that the number of failures is strongly and positively correlated with the length of individual belt sections. The density of extent can be expressed using Equations (3) and (4).

$$E_{Xdensity} = \frac{E_X}{L}, \tag{3}$$

$$E_{Ydensity} = \frac{E_Y}{L}, \tag{4}$$

where:

- $E_{Xdensity}$—density of longitudinal extent;
- $E_{Ydensity}$—density of transverse extent;
- $L$—length of the section in meters.

Therefore, the extent density is a parameter with the dimension of $\frac{mm}{m}$. The obtained values are dimensionally meaningful and easy to interpret, providing practical insights.

In the case of belt conveyor analysis, extent takes into account not only the presence of damage but also its size, such as length or width. It is a one-dimensional parameter

that provides a more comprehensive measure than the number of failures alone because it takes on different values depending on the size of the damage signal. This parameter allows for the distinction between small and large failures. Longitudinal extent refers to the length of the damage along the axis of the belt. For example, if a damage has a length of 30 mm, its longitudinal extent would be 30, indicating that the damage extends over a 30 mm section along the belt. Transverse extent is measured according to the number of measurement channels recording the damage (with channels spaced 25 mm apart), so it changes with a larger increment compared to longitudinal extent. Damage with a transverse extent of 2 channels means that the damage occupies two adjacent measurement channels, approximately 50 mm; thus, the value of the transverse extent parameter would be 50. Extent of the damage allows for a more comprehensive assessment of the belt's technical condition. The cumulative size of failures (their extent) provides different information than their number. The number or density of failures, as a relative measure (total number divided by the length of the belt section), does not provide an image of their size. Therefore, the analysis of longitudinal and transverse extent complements the picture by providing information about their size in both directions. As a result, diagnostic studies of belt conveyors can more precisely monitor and assess the belt's condition. This is particularly important as it enables early detection and response to serious failures that can lead to failures or operational issues. Figure 9 illustrates an example of calculating longitudinal and transverse extent of failures based on measurement signals.

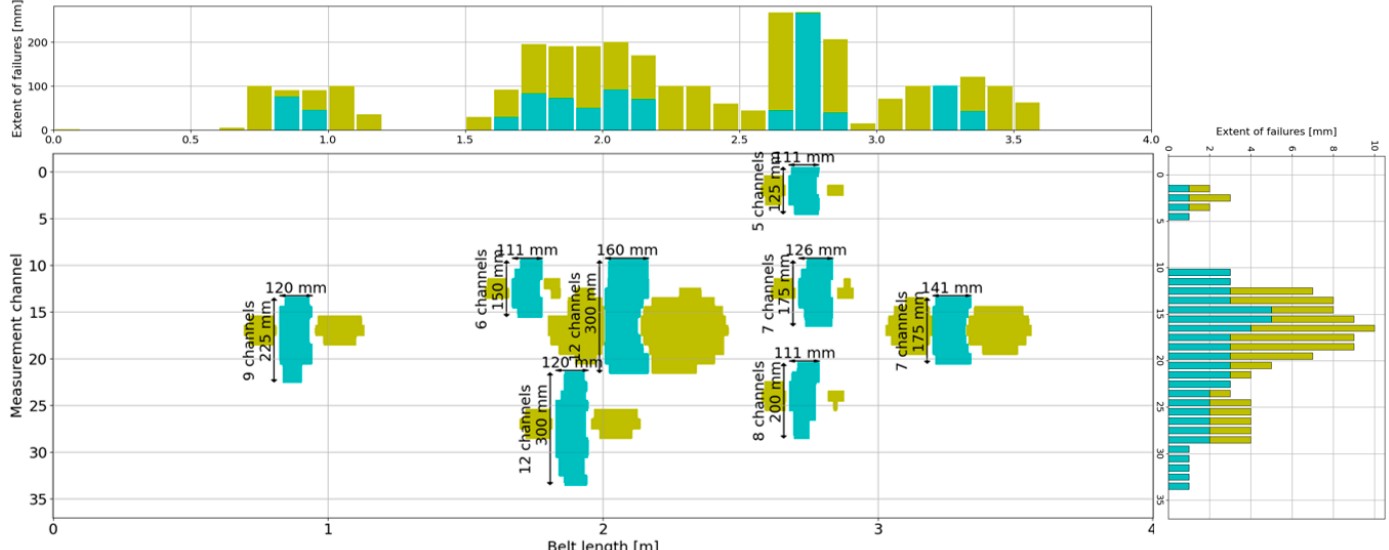

**Figure 9.** Method of calculating damage extent and histograms of longitudinal and transverse extent.

In the presented diagram, longitudinal and transverse extent can be calculated using Equations (5)–(8).

$$E_X = 120 + 111 + 120 + 160 + 111 + 111 + 126 + 141 = 1000 \text{ mm}, \tag{5}$$

$$E_Y = 225 + 150 + 300 + 300 + 125 + 200 + 175 + 175 = 1650 \text{ mm}, \tag{6}$$

$$E_{Xdensity} = \frac{1000}{4} = 250 \, \frac{\text{mm}}{\text{m}}, \tag{7}$$

$$E_{Ydensity} = \frac{1650}{4} = 275 \, \frac{\text{mm}}{\text{m}}, \tag{8}$$

The image presented in Figure 9 illustrates a section of the belt as well as Formulas (1)–(8) visualizing the method for calculating transverse (and longitudinal) extent for a belt section (in this example, a 4 m section with 37 measurement channels). The histograms depicting the frequency of failures or extent of failures are constructed using data from the entire belt loop.

Figure 10 presents an example of the damage extent graph for a belt, whose histogram of damage distribution across the belt is shown in Figure 8.

**Figure 10.** Example histogram of transverse extent of failures (conveyor A).

The graph representing the distribution of longitudinal extent ($E_X$) on each measurement channel is smoother because the length and sum of failures are real numbers, while the number of failures is a natural number. The nature of the distribution (its trimodality) is preserved in both graphs, and the chosen units may affect the scale of the axis. If the belt loop on the examined conveyor consists of multiple sections with different lengths (between connections) and different operating times (due to replacement processes [28]), it is best to use relative measures converted to meters of section length to describe the state of failures. Instead of providing the total number of failures on a given channel, it is better to provide the density of failures, which is the sum of failures ($N$) divided by the length of the entire loop ($L_{belt-loop}$) or the length of the analyzed section ($L_{belt-section}$) when comparing belt sections with each other (unit: number/meter). In the case of longitudinal or transverse extent, it is also valuable to analyze relative measures, such as the density of extent (Equations (3) and (4)), which indicate the extent value per meter of the belt (loop/section).

The article presents the results of measurements carried out on a series of belt conveyors operating in industrial conditions in Poland (in a lignite coal mine, limestone mine, and bituminous coal mine). The conveyors differed in terms of length, belt speed, strength, number of belt sections in the loop, and the type of material being transported. In each of the conducted studies, the sensitivity threshold of the measurement system was appropriately selected based on many years of experience in conducting measurements using the DiagBelt system [23,29]. Table 1 provides an overview of the collected information regarding the examined belt conveyors.

Figures 11 and 12 depict the method of mounting the magnet and measuring heads of the DiagBelt+ diagnostic system during measurements.

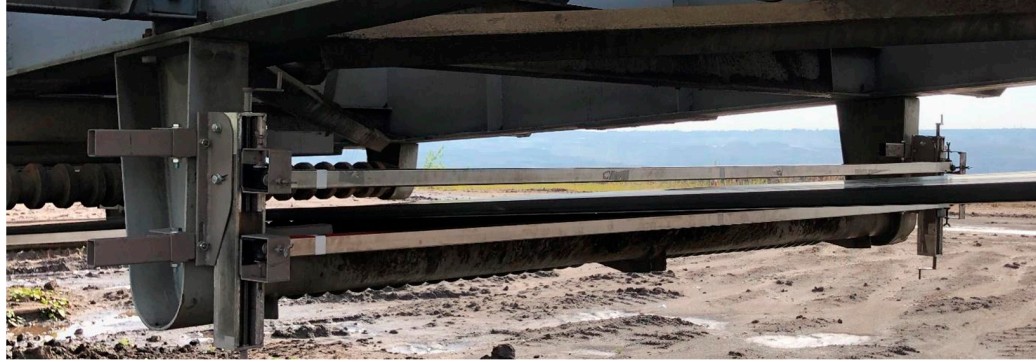

**Figure 11.** Two magnet heads attached to the conveyor structure.

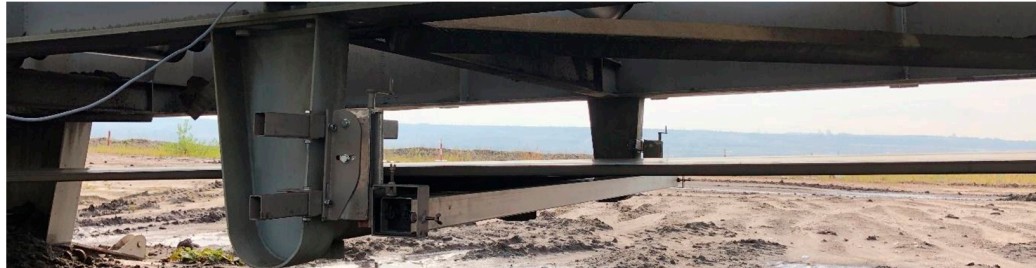

**Figure 12.** Measurement head placed on stands below the lower belt strand of the conveyor.

**Table 1.** Information about the examined conveyor.

| Conveyor | Length [m] | Transported Material | Belt Width [m] | Strength [kN] | Speed [m/s] | Number of Belt Sections |
|---|---|---|---|---|---|---|
| A | 2021 | overburden | 2.25 | 3150 | 6 | 11 |
| B | 2434 | overburden | 2.25 | 3150 | 6 | 11 |
| C | 2367 | overburden-brown coal | 2.25 | 3150 | 6 | 14 |
| D | 2974 | overburden | 1.8 | 3150 | 6 | 23 |
| E | 2214 | overburden | 2.25 | 3150 | 6 | 12 |
| F | 1881 | overburden | 2.25 | 3150 | 6 | 11 |
| G | 5189 | hard coal | 1.6 | 1600 | 2.1 | 14 |

## 3. Results

The material transported by the conveyor belt falls onto it at the loading point and moves along with the belt to the discharge point. The energy of the falling particles exceeds the critical value of energy that causes damage to the core, and the cords may be damaged [9,30–32]. The conveyor belt laid in a trough at the discharge point consists of three areas: two inclined sections and one flat section in the central part (Figure 13). The unsupported parts without idlers are small areas with a curvature connecting the inclined area with the flat section of the trough. When the bulk material falls onto the belt, it usually collides with one of these "flat" sections. At the moment of impact, the largest particles of the material may have energy exceeding the critical value, which can lead to belt penetration and damage to the cords in the core. The component of the impact force vector perpendicular to the inclined section of the core is smaller than in the case of impact on the flat section of the trough. Therefore, the same particle is less likely to damage the lateral parts of the belt compared to the central parts, precisely because of the uneven distribution of the component forces. A particle that failures the side part of the belt will need to have higher energy (compared to the central part) in order for the component force perpendicular to the belt surface to damage the cord in the core. The tangential component of the force (shearing force) can accelerate the processes of belt cover wear in those areas where the material is not only accelerated in the direction of belt movement but also shifted laterally due to the presence of this component. As a result of this variation and sometimes improper design of the loading point (causing uneven distribution of particle impacts on the belt), the distribution of damage will also be uneven. The observed trimodality of damage distribution (Figures 8 and 10) may be the result of loading a troughed belt. The optimization of the loading point design and belt arrangement at the loading point is an excellent research task for simulations using the Discrete Element Method (DEM) [33–38]. Diagnostics can only confirm the predicted load distribution obtained from the simulations. The selection of the inclination angle, the proportions of the inclined and flat sections, the use of energy-absorbing devices, and the reduction in speed differences between the belt and falling particles can contribute to eliminating local damage hotspots and achieving an even distribution of impacts and damage. Consequently, this can prolong the belt's lifespan (its operational time) and reduce transportation costs. It also leads to a reduction in resistance at the loading point and energy savings required to overcome them.

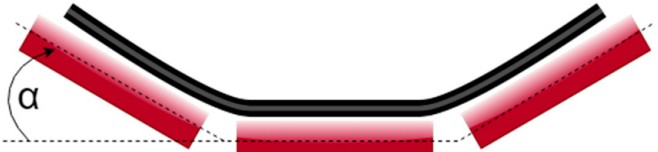

**Figure 13.** Conveyor belt operating in a trough—schematic.

The three modalities of damage distribution mean that there are three main areas where damage occurs. The central part of the belt, which should receive the highest energy impact, is the most damaged. This area experiences more frequent damage because the material has the highest probability of hitting it. In the bending areas of the belt, there is a noticeable decrease in the damage distribution. Although the trough theoretically provides a space for the material to settle, in practice, it is less likely that the material will hit precisely that spot. Factors such as material velocity, impact angle, and irregularities in the process can cause the material to strike closer to the "flat" parts of the belt than in the bending area, leading to an asymmetry in the damage distribution.

The results of the research regarding the distribution of core failures in the case of belt conveyors working in troughs are significant for assessing the technical condition of the belts and developing maintenance strategies. Analyzing the trimodal distribution of failures allows identifying the areas most vulnerable to damage and considering their asymmetric nature. This enables focusing on those areas during inspections, repairs, and maintenance of the conveyor belt.

A regularly damaged conveyor belt exhibits a symmetrical, trimodal distribution of damage density (or extent of damage). An example of such damage distribution is shown in Figures 14 and 15.

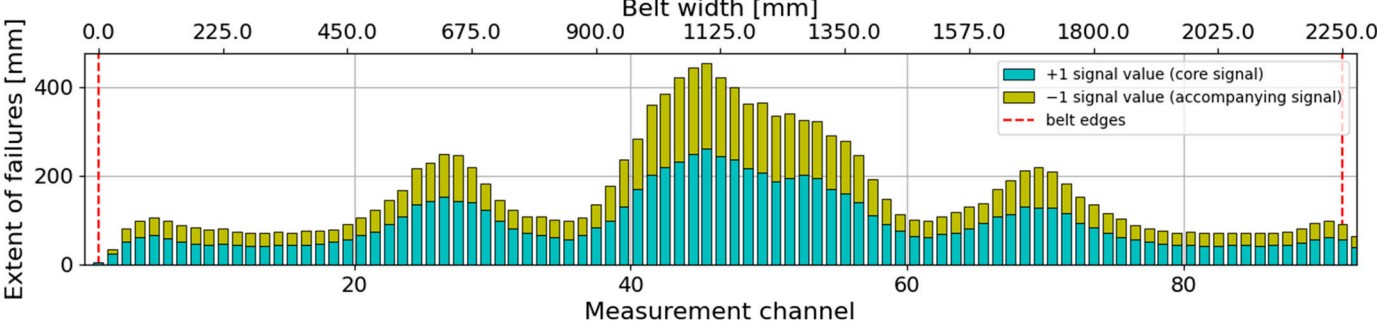

**Figure 14.** Histogram of transverse extent of failures—Conveyor B.

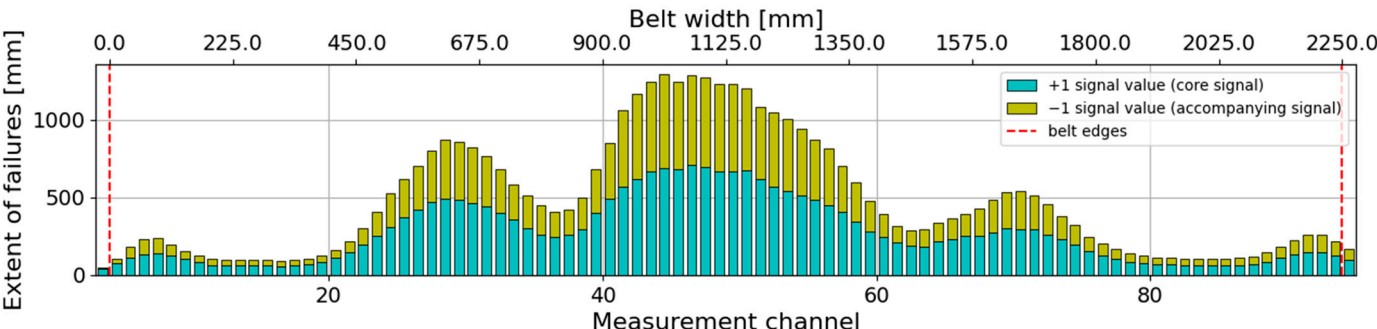

**Figure 15.** Histogram of transverse extent of failures—Conveyor C.

The purpose of the components used in transfer points is to properly direct the material stream onto the conveyor. They also ensure the correct positioning of the material on the receiving conveyor belt and prevent it from spilling beyond the belt. Studies on the reliability of conveyor systems have shown that approximately 30% of all failures occur at the material transfer point onto the conveyor [39]. Therefore, the durability of conveyor belts largely depends on how the issues related to material transfer are addressed. Poorly

designed transfer points can lead to operational interruptions, such as blockages in the return station of the conveyor due to material overflow. Additionally, the number and type of belt failures are largely dependent on the height of material discharge (the higher the material is discharged from, the greater its energy); thus, efforts should be made to minimize this height [39].

The equipment used in material transfer points includes impact bars, which focus the material discharged from the transfer drum into a compact stream and direct it in the desired direction; chute liners, which provide the falling material stream with the required velocity and direction; and loading chutes, which shape the material onto the receiving conveyor belt and prevent it from spilling beyond the belt. Figure 16 presents a schematic cross-section of a loading chute depositing material onto the conveyor belt. When the material stream is spread across the full width of the belt, it is possible for failures to occur in a trimodal distribution (Figures 14 and 15).

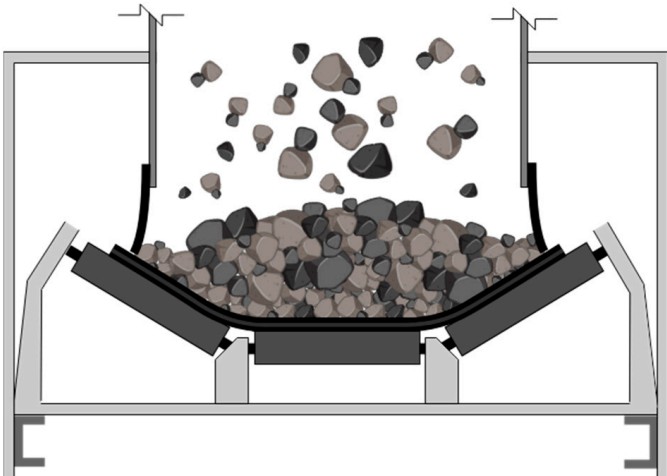

**Figure 16.** Overview diagram of the receiving conveyor with belt evenly loaded on its cross profile in the chute.

Monitoring and controlling the size and location of failures across the cross-section of a conveyor belt can provide valuable information about the load distribution on the belt. Through precise monitoring and analysis of damage locations, it is possible to identify areas where the most frequent and largest core failures occur. These areas are likely to be subjected to material stream impact at loading points (on transfer chutes). This can also be investigated using other methods, such as observing the material stream at those locations, placing special mats with sensors to record impacts from falling material and particles, or using simulation methods to analyze the process in a virtual world that replicates real conditions, including the size distribution of particles, the geometry of the loading chute, the receiving area, and a model of the conveyor belt onto which the material is transported. All of these methods require separate and costly field studies or the development of complex computer models. The simplest method, however, is to examine the condition of the belt in order to effectively manage its operation due to its high cost and high susceptibility to failures. Belt condition monitoring may also be necessary to verify the results of other studies, especially the results of DEM simulations. Virtual models require calibration and verification, and observing the effects of material stream impacts on the belt (distribution, number, and size of failures) serves as a very good method for validating the accuracy of these complex models [40–45].

However, when the falling material stream is concentrated only in the central part of the belt (Figure 17), the flat sections of the belt located on the sides are not exposed to impact energy, resulting in significantly fewer failures in those areas. The distribution of transverse damage intensity remains concentrated around the axis point of the belt, as depicted in Figure 18.

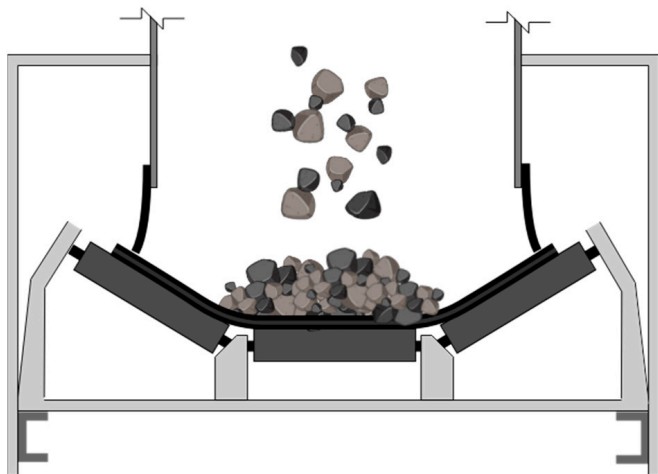

**Figure 17.** Overview diagram of the receiving conveyor with the belt centrally loaded in the chute.

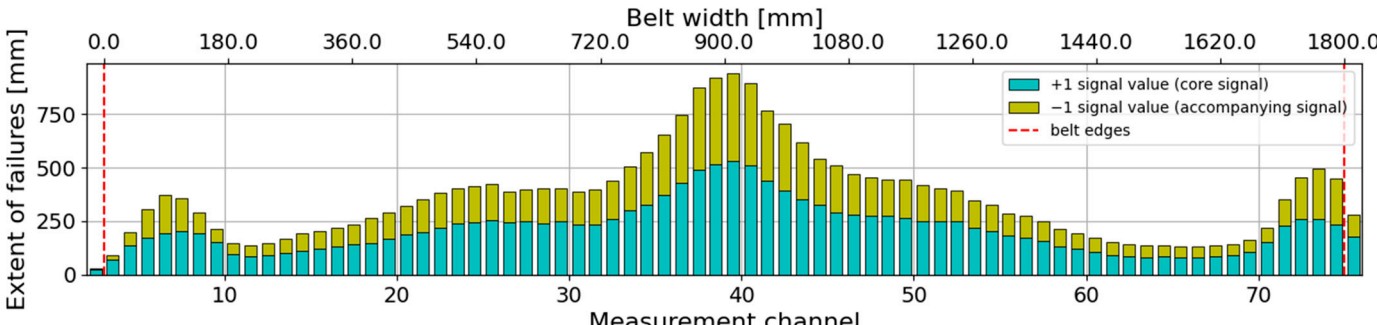

**Figure 18.** Histogram of transverse extent of failures—Conveyor D.

To ensure optimal functioning of the conveyor and optimal wear of its components, it is important to load the material onto the conveyor belt at its center. Non-centralized distribution of the conveyed load can lead to numerous maintenance and operational issues. When the material is not evenly distributed on the belt but forms a heap against one of the side guards (Figure 19), the belt can be displaced transversely. When the belt is not centrally loaded, the center of gravity of the load shifts as the material tries to find the center of the guiding idlers. This causes the belt to deviate, pushing it towards its less loaded side. This displacement poses challenges in belt alignment and can result in material spillage beyond the edge of the belt at the transfer point.

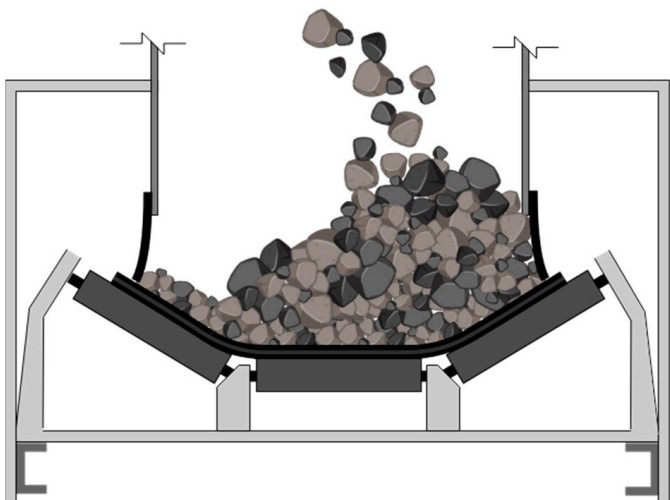

**Figure 19.** Overview diagram of the receiving conveyor with the belt asymmetrically loaded in the chute.

Even without belt displacement, the material falling onto one side of the belt causes excessive damage to that side. Such uneven loading of the belt at the transfer point can manifest in an inspection of the belt's core condition, where the distribution of failures across the transverse section of the belt remains trimodal but is no longer symmetrical. An example of a core damaged in this way is shown in Figures 20 and 21.

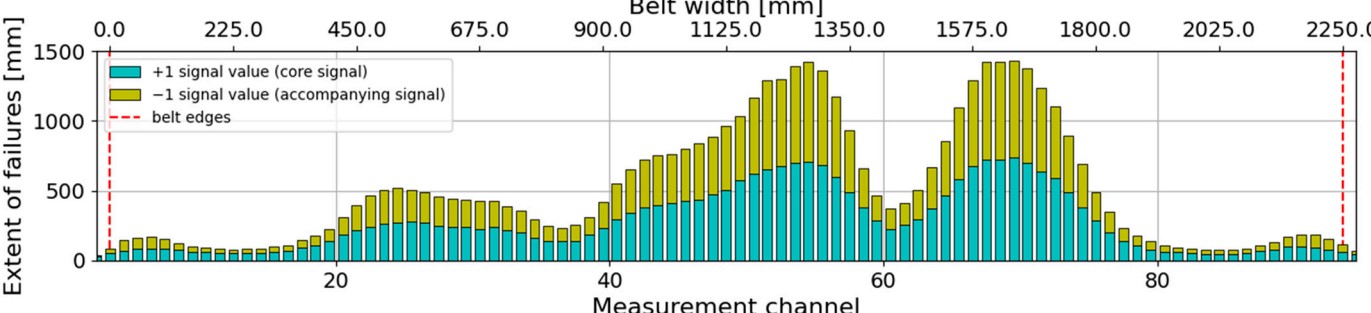

**Figure 20.** Histogram of transverse extent of failures—Conveyor E.

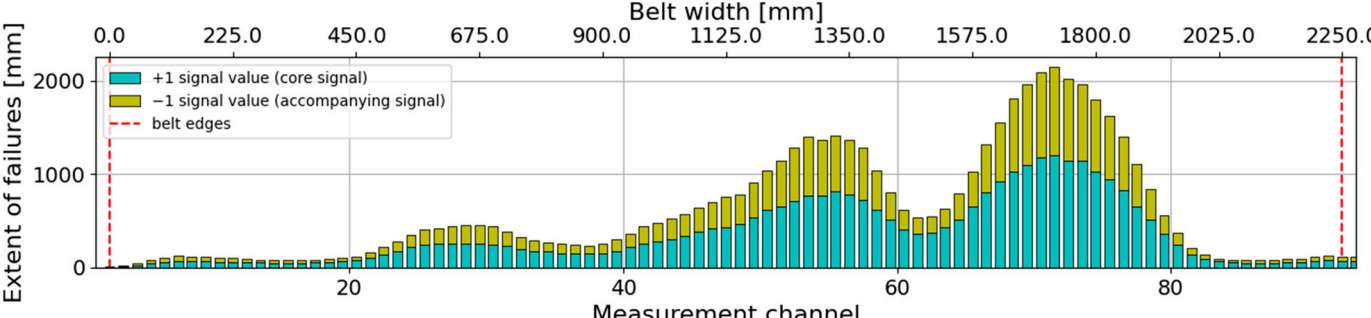

**Figure 21.** Histogram of transverse extent of failures—Conveyor F.

When conveyors operate in a straight line, without a change in material direction at the transfer point, achieving a centralized flow of the load is relatively straightforward. In such cases, it is easy to place the material precisely at the center of the receiving belt, ensuring even distribution of the load and minimizing material spillage issues. However, more commonly, situations arise where a change in direction is required, which is why transfer points are installed. There are various strategies and components that can be employed to redirect material flow at non-linear transfer points. However, there are certain drawbacks associated with angled transfer points. It becomes more challenging to maintain the proper angle, trajectory, and flow velocity at the transfer point.

Sometimes, deflectors are installed inside the chute to absorb impact energy and minimize wear. The deflector's purpose is to change the direction of material flow and direct it to the appropriate location. Curved sections are often used at the bottom of the chute to control the spread of material. These curves guide the material stream at the proper angle, reducing scattering and material degradation. In some cases, deflectors are used to change the direction of material flow. The deflector is placed at the end of the conveyor or near the transfer point between conveyors. Its task is to redirect the material in a different direction, providing control over its flow.

The rubber skirt (black, perpendicular to the belt, rubber element in Figures 16, 17 and 19) is a component of the loading chute and is used in conveyor systems to control material scattering at the loading point. It is installed on the sides of the loading chute, perpendicular to the conveyor belt. The main purpose of using the rubber skirt in the loading chute is to minimize material leakage from the conveyor. The skirt creates a tight seal between the conveyor belt and the sides of the loading chute, preventing material from falling out and scattering sideways. It acts as a barrier that keeps the material on the belt and prevents it from escaping. However, if the skirt is pressed too tightly against the belt, it can wear out the belt and reduce the thickness of the belt covers at the edges, sometimes even exposing

the cords. The reduced thickness of the cover supports the formation of core damage in that area, and in the case of cord exposure, it can even enhance corrosion. In such a situation, a slight increase in the values in the transverse distribution of extent defects in the belt core may be noticeable around the edges (Figures 15, 18, 20 and 21). However, when the pressure of the skirt against the belt is not excessive, there is no increase in the values (Figure 22).

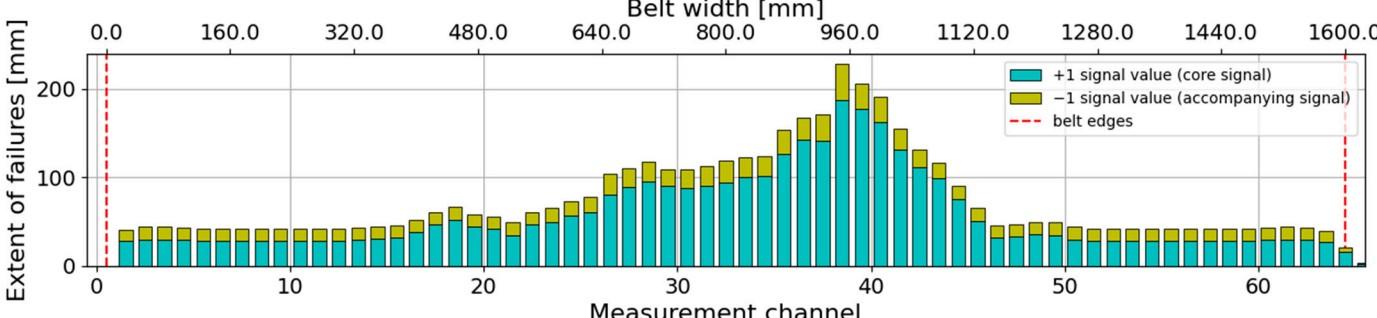

**Figure 22.** Histogram of transverse extent of failures—Conveyor G.

In the presented result of the conveyor G examination in Figure 22, although the extent defects do not increase at the edges, an uneven distribution is visible. As mentioned earlier, this can be a result of uneven loading of the conveyor, but it can also be caused by improper tension in the belt. Due to building settling or other factors, sometimes a part of the conveyor, such as the tensioning drum, may rotate slightly. Such unplanned rotation can lead to uneven tension in the conveyor belt. This is often visible in visual inspections because excessively tensioned cords in the steel core are visible on the belt cover (an example is shown in Figure 23).

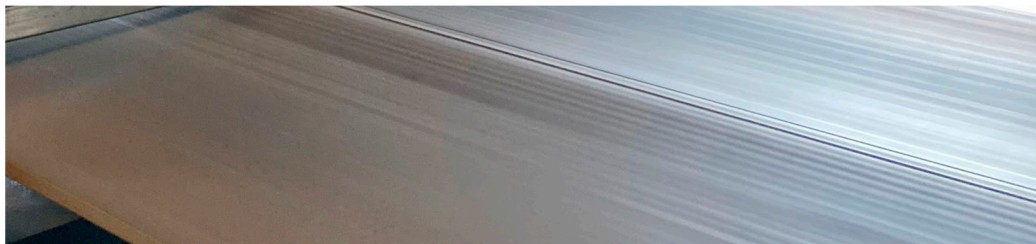

**Figure 23.** Example of visible cords in an excessively tensioned belt.

## 4. Conclusions

Thanks to the diagnostics of the conveyor belt core using the DiagBelt diagnostic system and a thorough analysis of the obtained results, it is possible to effectively identify many faults and problems in the conveyor's design. The analysis of the obtained results allows for the assessment of the technical condition of the conveyor belt core and the identification of areas requiring attention. Detecting faults at an early stage helps to avoid more serious damage, failures, and downtime of the conveyor. This enables effective planning of maintenance, upkeep, and repairs. It is crucial for ensuring the reliability, efficiency, and safety of the conveyor's operation, which translates into the efficiency of production processes and the reduction in costs associated with failures.

Furthermore, the DiagBelt diagnostic system is developed and continually improved by the authors of this article. Ongoing efforts are dedicated to enhancing its capabilities and incorporating new features, such as alerts in case of exceeding predefined damage thresholds. This advancement supports the installation of the device for continuous operation. Additionally, the signals indicating failures undergo research as part of doctoral studies to determine the time to the next failure using artificial intelligence methods. The ongoing research also includes the classification and recognition of signals. The outcomes of these studies will be disseminated through subsequent articles, contributing to the continual refinement and advancement of the DiagBelt diagnostic system.

The conclusions and information obtained through the DiagBelt system and result analysis provide valuable knowledge for technical and engineering staff, enabling them to take appropriate corrective and optimization actions to maintain the conveyor in optimal technical condition.

If failures occur in a specific area across the cross-section of the belt on each section of the loop, the belt may be unevenly loaded by the falling material, leading to excessive wear and reduced durability in that particular area. The decision to dismantle the belt is often made when the core of the belt is exposed. This occurs later when the belt is evenly worn across the entire cross-section. The impact energy and friction are distributed over a larger surface area, which prolongs the process of core exposure. However, when the core destruction and friction processes (e.g., accelerated material at the loading point) are concentrated in selected areas (where local maximums of failures occur), the time to expose the core is significantly shortened.

Monitoring and analyzing the size and distribution of failures across the cross-section of the conveyor belt allows for the identification of areas that require special attention and potential adjustments to the material transport process through modifications of loading points and conveyor parameters (belt speed, incline angle, or supporting methods at the loading points). The goal is to achieve an even distribution of load on the belt. Sometimes, when the distribution is uneven, a better solution may be to temporarily redirect the material flow to the least worn areas in order to extend the belt's lifespan by reducing the load and wear rate in the most damaged sections of the belt's cross-section. After replacing the belt with a new one, it is important to ensure an even distribution of failures to prolong its service life.

In the case of conveyor belts operating in industrial conditions, the belt loop can consist of sections with varying degrees of wear. This is because not all belt sections are always installed new, and they may not operate for the same duration. Some sections may be transferred from other conveyors (e.g., decommissioned, dismantled, or relocated). As a result, it is possible that different sections of the belt loop will have varying degrees of wear or even a different wear distribution across the cross-section (if they were transferred from other conveyors). However, if the cause of failures in a specific location of the cross-section is due to an improper design of the loading point and conveyor, new failures will continue to occur in the same locations on the belt.

Irregularities in the conveyor's design, such as improper idler alignment, incorrect belt tension, uneven support, improper belt tracking, or an ill-matched discharge angle, can result in excessive wear, cuts, cracks, or other failures in specific locations of the belt loop. Even after replacing a damaged belt section with a new one, if the underlying cause of the improper operation is not addressed, the failures will continue to occur in the same points on the cross-section. As the thickness of the conveyor belt covers decreases due to wear, the critical energy of falling particles causing core failures will also decrease. Consequently, the same stream of material will cause more failures, and the rate of their occurrence will increase. The fraction of particles posing a threat to the belt (those with sufficient energy to cause failures) will increase as the thickness of the top cover decreases.

If any abnormalities are observed, such as an asymmetric distribution of failures (Figures 20 and 21), which may result from uneven material feeding onto the belt (Figure 19), corrective actions should be taken. Failure to take action will likely lead to accelerated wear of the belt in the area of the highest damage concentration, ultimately reducing its durability. Addressing this issue and reversing the trend would require an analysis of the material feeding process at the loading point and potential design changes to the loading chute to ensure that the material starts falling onto the opposite, less damaged side of the belt. The consequences of changes in the transfer geometry can be examined by modeling the material feeding process in DEM simulators.

Storing the acquired measurement data in a database is of great importance for effective monitoring of the belt conveyor's condition and evaluation of the effectiveness of repair actions. Recording and analyzing measurement data enable the creation of his-

torical records of the conveyor's condition. This allows for tracking changes and extent increments over time, which is extremely useful for evaluating the effectiveness of repair actions taken. Comparing historical data with current data allows for an assessment of whether the actions taken have produced the desired effect and eliminated the cause, and whether the identified problem was the sole factor contributing to uneven wear of the belt conveyor's core. This way, trends and potential deviations from the norm can be monitored, potential problems can be identified, and appropriate corrective actions can be taken. Access to the complete history of the conveyor's technical condition facilitates the planning of maintenance, upkeep, and modernization. Knowledge of previous states and the effectiveness of previous repairs provides a valuable reference point and basis for making decisions regarding further actions. Therefore, maintaining a database system with comprehensive measurement data is an essential element of effective belt conveyor management. It enables the effective monitoring, assessment, and optimization of the conveyor's condition, resulting in the increased reliability, efficiency, and safety of its operation.

**Author Contributions:** Conceptualization, L.J.; methodology, L.J. and R.B.; software, L.J., A.R. and A.K.-B.; validation, L.J. and R.B.; formal analysis, L.J. and A.R.; investigation, R.B. and A.K.-B.; resources, R.B. and A.K.-B.; data curation, R.B.; writing—original draft preparation, L.J. and A.R.; writing—review and editing, A.K.-B.; visualization, L.J. and A.R.; supervision, R.B. and L.J.; funding acquisition, R.B. All authors have read and agreed to the published version of the manuscript.

**Funding:** This research received no external funding.

**Data Availability Statement:** The data are not publicly available due to restrictions, as they contain information that could compromise the privacy of research participants.

**Acknowledgments:** The authors would like to thank the brown coal mine Belchatow for providing conveyor belts for tests.

**Conflicts of Interest:** The authors declare no conflict of interest.

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
