# Peer review of "Transverse Profiles of Belt Core Damage in the Analysis of the Correct Loading and Operation of Conveyors"

_minerals, doi:10.3390/min13121520_

Round 1

Reviewer 1 Report

Comments and Suggestions for Authors

Dear Authors

I would like to thank you for providing quantitative information on your system called "Diagbelt". It presents accurate and precise information on possible problems regarding belt conveyors. On the other hand, the authors only stated and presented the strengths of the system "Diagbelt" in the article. They can also put forward and suggest additional ideas and some alternatives to improve their system before completing the article. For example, in case of the presence of excessive loading on the belt conveyor, can Diagbelt measure the variations in current in the motor propelling the whole system? Similarly, can we add or adapt some start-stop transmissions in the conveyor belt to reduce/minimize the failures in/on the related components? From this point of view, I guess, Diagbelt can provide promising results regarding the detection and protection of possible problems in conveyor belts.

In order to improve the quality of the paper:

- I recommend you improve the language and flow of the article.

-For the sake of clarity, please give some legends to the Figures (for example in Figure 11: Two colours may represent different types of failures ( I it is not clear, please give a legend for each color in these figures). 

- In Figure 5, a flat conveyor belt can be seen at your laboratory. However, during the analysis of the conveyor belts for example in Figure 17-18, some are carrying idles with a through angle of 35 degrees can be seen. Please verify whether your system at the laboratory can be adaptive to carrying idlers with varying through angles. 

-Kindly note that there should be no citations and references in the conclusion section. Please reconsider your conclusion section. In a typical conclusion, the findings, main points declared in the article and some suggestions can be mentioned.

- Also please give some additional information on the analyzed conveyor belts such as belt weight (kg/m) and the thickness of the belt (mm), if possible.

I hope the above recommendations would help improve your article.

Comments on the Quality of English Language

The Quality of English can be improved by detailed proofreading. 

Author Response

Dear Reviewer,

We sincerely appreciate your valuable reviews and insightful analysis of our manuscript.
The manuscript has been revised in accordance with your comments. Below, we have addressed your suggestions – our responses are presented in green.

Thank you for your time and dedication to improving the quality of our work.

Best regards,    

Authors

Dear Authors

I would like to thank you for providing quantitative information on your system called "Diagbelt". It presents accurate and precise information on possible problems regarding belt conveyors. On the other hand, the authors only stated and presented the strengths of the system "Diagbelt" in the article. They can also put forward and suggest additional ideas and some alternatives to improve their system before completing the article. For example, in case of the presence of excessive loading on the belt conveyor, can Diagbelt measure the variations in current in the motor propelling the whole system? Similarly, can we add or adapt some start-stop transmissions in the conveyor belt to reduce/minimize the failures in/on the related components? From this point of view, I guess, Diagbelt can provide promising results regarding the detection and protection of possible problems in conveyor belts.

Thank you for your feedback. The DiagBelt+ diagnostic system is designed for assessing the technical condition of the steel cords in the core of the St belt. It does not have the capability to monitor the current in the motor driving the belt. However, the installation of a measuring system permanently, just behind the loading point, could indeed be part of a system that stops the conveyor when a serious failure is detected. Most failures to the steel cords in the belt core develop gradually, and the use of cyclic scanning allows for monitoring the rate of damage growth, enabling predictive repairs. We have described such studies in other works, for example: [19, 20, 22]. Purchasing a single measurement head and cyclically scanning the, core's condition is much more cost-effective and allows for the management of multiple conveyors simultaneously. The mine from which the data originates has recently purchased one measurement head, using it to scan the existing belts and make decisions about directing belts for dismantling.

The pertinent information has been incorporated into the conclusion section.

In order to improve the quality of the paper:

  • I recommend you improve the language and flow of the article.

Implementing the correction is challenging as it pertains to the entirety of the text. Nevertheless, the second reviewer emphasized that the text is well-written and easy to read. We have changed few words and sentences, add explanations, as well as moved text fragments from conclusions to another location.

  • For the sake of clarity, please give some legends to the Figures (for example in Figure 11: Two colours may represent different types of failures ( I it is not clear, please give a legend for each color in these figures).

The legend has been added. The various colours on the charts represent the point clouds with different polarizations of magnetic field. The blue colour is associated with the presence of damage, while the yellow colour (accompanying signal) occurs, when the damage is significant. Therefore, both the blue signal and the yellow signal are important. An example of the damage signal is presented in Fig. 6 or Fig. 10)

The description of the damage signal and its categorisation into core signals and accompanying signals can be found in lines 131-139 of the manuscript. Analysing the relationships between the magnitudes of the core and accompanying signals alone will be valuable for further analysis.

  • In Figure 5, a flat conveyor belt can be seen at your laboratory. However, during the analysis of the conveyor belts for example in Figure 17-18, some are carrying idles with a through angle of 35 degrees can be seen. Please verify whether your system at the laboratory can be adaptive to carrying idlers with varying through angles.

The belt depicted in Fig 5 is too narrow to fit into the trough (the side rollers protrude almost entirely beyond the belt). Currently, in the laboratory, we have another wider belt that has been laid in the trough; however, the results of the studies using this belt have not been published thus far. However, in the laboratory, we have the ability to set the idlers at an angle. However, the DiagBelt+ measurement system is installed on a flat section of the conveyor belt's path (most commonly on the lower strand) to maintain the same distance from the measurement strip to the belt (see Figures 12 and 13). The schematic diagram of the loading chute presented in Figures 17-18 illustrates the arrangement of the belt at the material receiving point, where the probability of core damage is highest (due to the energy of falling rocks). The laboratory conveyor is used for testing diagnostic equipment and does not transport material.

  • Kindly note that there should be no citations and references in the conclusion section. Please reconsider your conclusion section. In a typical conclusion, the findings, main points declared in the article and some suggestions can be mentioned.

It has been corrected.

  • Also please give some additional information on the analyzed conveyor belts such as belt weight (kg/m) and the thickness of the belt (mm), if possible.
  • Table 1 contains comprehensive information about the belts in our possession. Previous research indicates that, on average, the weight of belts installed in the mine from which the examined belts originate is approximately 42-45 kg/m, with a thickness of 14 mm for the carrying cover and 7 mm for the running cover. However, due to uncertainty regarding whether all the belts described in the article share these exact parameters, we have chosen not to include this information in the table.

Reviewer 2 Report

Comments and Suggestions for Authors

The paper is well-written and easy to read. It presents an interesting analysis of transverse damage profiles of conveyor belts in the mining industry. Some remarks:

* The paper would benefit from more detailed explanations of the algorithms for the data processing part of the DiagBelt system, i.e., how you arrived at the presented histograms.

* On a related note, could you explain a bit more the nature of the "accompanying signals", especially the distinction with the "core signal". Maybe a visualization of the original measurement data could help?

* You mention in the abstract that seven conveyors were examined in total, and the histogram plots also show seven different conveyors, however, Table 1 lists eight different conveyor belts. Does conveyor H have any relevance here?

* Could you perhaps also include some details on the loading chutes used for these different conveyors? Since this is an important aspect for belt damages and extent profiles, as you argue later in the paper.

* The measured "extent of failures" gives an indication, however, by itself is not sufficient to predict, e.g., remaining time before maintenance, in order to support maintenance planning, repairs etc. This would be an interesting future study.

* Some sections of the text could be streamlined to make them more concise. There are some repetitions of sentences and statements. E.g., on Page 9, Page 11, and in the conclusion.

* In Figure 24, it is difficult to identify these cords as described. Perhaps consider using a different picture?

* There is a typo in line 443: "Figure 21" --> should be "Figure 23".

Author Response

Dear Reviewer,

We sincerely appreciate your valuable reviews and insightful analysis of our manuscript. The manuscript has been revised in accordance with your comments. Below, we have addressed your suggestions – our responses are presented in green.

Thank you for your time and dedication to improving the quality of our work.

Best regards,    

Authors

The paper is well-written and easy to read. It presents an interesting analysis of transverse damage profiles of conveyor belts in the mining industry. Some remarks:

  • The paper would benefit from more detailed explanations of the algorithms for the data processing part of the DiagBelt system, i.e., how you arrived at the presented histograms.

It has been added.

  • On a related note, could you explain a bit more the nature of the "accompanying signals", especially the distinction with the "core signal". Maybe a visualization of the original measurement data could help?

The various colors on the charts represents the point clouds with different polarizations of magnetic field. The blue color is associated with the presence of damage, while the yellow color (accompanying signal) occurs, when the damage is significant. Therefore, both the blue signal and the yellow signal ale important. An example of the damage signal is presented in Fig. 6 or Fig. 10)

The description of the damage signal and its categorization into core signals and accompanying signals can be found in lines 131-139 of the manuscript. Analyzing the relationships between the magnitudes of core and accompanying signals alone will be valuable for further analysis.

The images presented in Figure 6 serve as a visualization of raw data (matrices of values -1, 0, 1) in the form of a two-dimensional image.

You mention in the abstract that seven conveyors were examined in total, and the histogram plots also show seven different conveyors, however, Table 1 lists eight different conveyor belts. Does conveyor H have any relevance here?

The conveyor H has been removed from the table because it was unnecessary.

  • Could you perhaps also include some details on the loading chutes used for these different conveyors? Since this is an important aspect for belt damages and extent profiles, as you argue later in the paper.

The mine did not grant permission for the presentation of detailed information regarding the loading chutes

  • The measured "extent of failures" gives an indication, however, by itself is not sufficient to predict, e.g., remaining time before maintenance, in order to support maintenance planning, repairs etc. This would be an interesting future study.

Yes, the remaining time before maintenance is predictable using some data from the DiagBelt+ system (not only the extent of failures) and it is our future work. We have added information in the Conclusion section.

  • Some sections of the text could be streamlined to make them more concise. There are some repetitions of sentences and statements. E.g., on Page 9, Page 11, and in the conclusion.
  • In Figure 24, it is difficult to identify these cords as described. Perhaps consider using
    a different picture?

The photo reveals a crack (at the centre of the belt's width) and parallel lines stemming from the belt's bulge and cable tensions due to excessively high belt tension. The belt shown in the picture has a relatively low strength (1600 kN/m), so the cords in the core of the belt also have a relatively small diameter.

  • There is a typo in line 443: "Figure 21" --> should be "Figure 23".

It has been corrected.

Reviewer 3 Report

Comments and Suggestions for Authors

Line 300: In the manuscript it is not necessary to include Figure  13. Figure 12 has sufficient explanatory value and it is duplicate.

Line 442: "In the presented result of the conveyor G examination in Figure 21", should be correctly in "Figure 23".

Line 637: Is reference 40 correctly cited - “Golka, K., Bolliger, G., Vasili, C. (2007). Belt Conveyors: Principles for Calculation and Design. Australia: K. Golka, G. Bolliger, . Vasili"?

In the introduction , the authors could state what are the differences in the monitoring of belt conveyors with with a textile core and conveyors with steel cords.

In conclusion, the authors could state in which direction their further research in the field of monitoring of conveyor belts will develop.

Author Response

Dear Reviewer,

We sincerely appreciate your valuable reviews and insightful analysis of our manuscript. The manuscript has been revised in accordance with your comments. Below, we have addressed your suggestions – our responses are presented in green.

Thank you for your time and dedication to improving the quality of our work.

Best regards,    

Authors

  • Line 300: In the manuscript it is not necessary to include Figure Figure 12 has sufficient explanatory value and it is duplicate.

Duplicating the charts is intended to illustrate that the method of damaging conveyors in the same mine is similar, confirming that a similar design of chutes leads to analogous damage distributions.

  • Line 442: "In the presented result of the conveyor G examination in Figure 21", should be correctly in "Figure 23".

It has been corrected.

  • Line 637: Is reference 40 correctly cited - “Golka, K., Bolliger, G., Vasili, C. (2007). Belt Conveyors: Principles for Calculation and Design. Australia: K. Golka, G. Bolliger, . Vasili"?

The correction has been made. Two of the editors are also the authors of the book.

  • In the introduction, the authors could state what are the differences in the monitoring of belt conveyors with with a textile core and conveyors with steel cords.

The subject of the study were St belts; we do not have equipment for assessing the technical condition of the core of textile belts

  • In conclusion, the authors could state in which direction their further research in the field of monitoring of conveyor belts will develop.

It has been added.